# Method for Classifying Behavior of Livestock on Fenced Temperate Rangeland in Northern China

**DOI:** 10.3390/s19235334

**Published:** 2019-12-03

**Authors:** Xiaowei Gou, Atsushi Tsunekawa, Fei Peng, Xueyong Zhao, Yulin Li, Jie Lian

**Affiliations:** 1The United Graduate School of Agricultural Sciences, Tottori University, 4-101 Koyama-Minami, Tottori 680-8553, Japan; gou_xiaowei@126.com; 2Arid Land Research Center, Tottori University, 1390 Hamasaka, Tottori 680-0001, Japan; tsunekawa@tottori-u.ac.jp; 3International Platform for Dryland Research and Education, Tottori University, 1390 Hamasaka, Tottori 680-0001, Japan; 4Key Laboratory of Desert and Desertification, Northwest Institute of Eco-Environment and Resources, Chinese Academy of Sciences, Lanzhou 73000, China; 5Naiman Desertification Research Station, Northwest Institute of Eco-Environment and Resources, Chinese Academy of Sciences, Tongliao 028300, China; zhaoxy@lzb.ac.cn (X.Z.); liyl@lzb.ac.cn (Y.L.); lianjeco@gmail.com (J.L.)

**Keywords:** livestock, behavior classification, GPS, accelerometer, Random Forest, Kappa coefficient, dryland

## Abstract

Different livestock behaviors have distinct effects on grassland degradation. However, because direct observation of livestock behavior is time- and labor-intensive, an automated methodology to classify livestock behavior according to animal position and posture is necessary. We applied the Random Forest algorithm to predict livestock behaviors in the Horqin Sand Land by using Global Positioning System (GPS) and tri-axis accelerometer data and then confirmed the results through field observations. The overall accuracy of GPS models was 85% to 90% when the time interval was greater than 300–800 s, which was approximated to the tri-axis model (96%) and GPS-tri models (96%). In the GPS model, the linear backward or forward distance were the most important determinants of behavior classification, and nongrazing was less than 30% when livestock travelled more than 30–50 m over a 5-min interval. For the tri-axis accelerometer model, the anteroposterior acceleration (–3 m/s^2^) of neck movement was the most accurate determinant of livestock behavior classification. Using instantaneous acceleration of livestock body movement more precisely classified livestock behaviors than did GPS location-based distance metrics. When a tri-axis model is unavailable, GPS models will yield sufficiently reliable classification accuracy when an appropriate time interval is defined.

## 1. Introduction

Drylands cover more than 41% of the Earth’s land area, and desertification directly affects more than 250 million people [1]. Overgrazing is considered to be the primary cause of land degradation [2]. Previous studies examining overgrazing of rangeland generally used the number of livestock in a given area as the grazing intensity; this practice assumes that livestock foraging is spatially distributed evenly and that all livestock behaviors have the same influence on the rangeland [3]. However, the livestock always shows patchy and selective grazing even in homogenous rangeland to minimize their activity range and to maximize energy use efficiency [4]. In fact, vegetation typically shows a mosaic distribution, whether induced by abiotic factors, such as elevation and slope, or by selective grazing, which aggravates the overuse of some areas of the grassland [5].

The spatial distribution of different behavioral activities was critical for understanding the effects of grazing on ecosystem function, growth, reproduction and survival, how to make efficient use of resources [6], and mechanisms for coping with environmental conditions [6]. In the grazing areas, the vegetation was significantly reduced by the selective foraging of livestock. Moreover, concentrated grazing depletes the soil of nutrients [7], thus promoting further degradation of grassland [8], whereas light grazing can improve plant diversity by restraining inherent inter and intra-specific competition [9]. In comparison, nongrazing behaviors, including resting and walking, trample plants and compact the soil surface in overused areas, and the cumulative deposition of excreta alters various physical properties of soil, including soil bulk density, aggregate stability, aggregate size distribution, and surface microrelief. Recovering rangeland from degradation due to nongrazing behaviors is considered more difficult than remediating the effects of concentrated grazing [10]. 

Accurately classifying different behaviors of livestock is necessary to understand rangeland degradation and to devise effective interventions to restore the degraded land. One such method involves applying several statistical [11] and deep-learning [12] models to collected data from accelerometers for classifying livestock behaviors, which have been developed by using large data sets placed on animals in managed grassland [13,14]. These accelerometers measure the instantaneous and independent local movement of animals’ legs, heads, or bodies, thus ensuring high accuracy of behavior classification [15,16,17,18]. However, accelerometers cannot provide information regarding the location of the livestock, which is crucial for identifying the spatial distribution of animals and grassland management. Another method is to use Global Positioning System (GPS) data and machine-learning algorithms to classify livestock behaviors [19]. Using the location records, the GPS data-based method can project the spatial distribution of various behaviors, which is crucial for herd management and the prevention of rangeland degradation. However, GPS data-based methods require an optimal time interval, during which metrics such as linear distance (d), cumulative distance (d), and turning angle (t) are calculated to predict behaviors [12]. To build models for predicting livestock movement, the time intervals for metric calculation have previously been selected empirically [19,20]. The optimal time interval for GPS data-based methods varies with the ecosystem, livestock species, topography, and spatial distribution of available resources to evaluate [21].

The Horqin Sandy Land in northern China has been seriously degraded since the mid-1980s, and various restoration countermeasures (e.g., fencing) have been introduced to restore the degraded land [22]. In Horqin Sandy Land, the average area of the fenced rangeland per household is approximately 15–30 ha [23]. Fencing limits the space, and thus the forage, available to animals and consequently might aggravate mosaic grazing in areas; in addition, dense walking along the fence might lead to mosaic degradation. The objectives of our study were to develop a method for classifying livestock behavior by using location information and to define the optimal time interval for a GPS data-based model for fenced rangeland.

## 2. Materials and Methods

The study was conducted in a fenced household pasture, which is located in the southwestern part (42°55′N, 120°42′E; altitude, ~360 m) of Horqin Sandy Land, China. The climate is temperate, semi-arid, continental, and monsoonal. Average annual precipitation is 360 mm, with an annual mean temperature of 6.4 °C. The minimal and maximal monthly mean temperatures are −13.1 °C in January and 23.7 °C in July, respectively.

The pasture was grazed by Simmental cattle from 1 July through 1 October, 2018 (three months). During our study, the rangeland area was 20.1 ha, and herd size was 13 cattle. The stocking rate was calculated in terms of the common method [24], which the value was 0.51 Animal Unit Months per hectare. The total grazing time was approximately 3 months yearly due to the implement of ‘suspending grazing’ policy by the local government, which was for preventing grassland degradation. The availability of forage in our study area was about 53 g/m^2^ in July and 243 g/m^2^ in August for enclosure rangeland [25]. The vegetation was composed mainly of herbage belonging to arid grassland types (*Pennisetum centrasiaticum, Cleistogenes squarrosa*), with some dwarf shrubs (*Artemisia oxycephala, Artemisia halodendron*).

### 2.1. Equipment and Animals

All 13 cattle in the pastured herd were fitted with GPS devices (catalog no. GT-600, i-gotU, Mobile Action Technology, Taipei, Taiwan) and tri-axis accelerometers (catalog no. UA-004-64, Hobo model, Onset, Bourne, MA, USA). GPS devices were attached on the neck only, whereas tri-axis accelerometers were placed on the neck, one leg, and the tail of each animal. The GPS device recorded cattle location at 50 s intervals throughout two consecutive days, after which the GPS devices were removed, recharged, and re-attached to the cattle; this process continued throughout the 10-d study period. The three-dimensional accelerometers recorded the anterior–posterior, transverse, and superior-inferior acceleration of livestock movement. The batteries of the tri-axis devices were able to record acceleration at 50 s resolution throughout the 10-day study period without needing to be recharged.

### 2.2. Observation of Livestock Behaviors

Classification and criteria for animal behavior followed the method of Ganskopp and Bohnert [12]. In the experiment period, one observer observed one cattle at two days. According to our observation, a herd of cattle behaved similarly in a group. Thus, the observed behavior can represent the behavior of the cattle. In each day, the observer kept tracking one randomly selected cattle. The direct visual behavioral observation was recorded continuously by one observer following one cattle at approximately 20 meters away from the cattle in consecutive two days (23 and 24 September 2018). The observer held a timer which is synchronized with the time of the GPS. The field observation of behaviors started from 9:00 local time. The time interval of the GPS to record each location is 50 s. The GPS will flash when recording the location of the cattle. When the GPS flashes, the observer will read the timing from the timer and record the cattle behavior. If the cattle were foraging with head down when the GPS recording the location, it is considered as grazing behavior. If the cattle were standing still, chewing, or walking it is considered as nongrazing behavior. In total, 9 hours and 539 behaviors were recorded; approximately 80% of activities were grazing behaviors, and the remaining 20% was the nongrazing activity. Detailed information regarding the behavior classification is given in Table 1.

### 2.3. Movement Metrics Derived from GPS and Tri-Axis Accelerometer Data

Coordinates of GPS device were converted from latitude/longitude form to a Universal Transverse Mercator (UTM) format to facilitate metrics of distances and turning angle [20]. Metrics related to distances cattle moved and the turning angle were derived to classify the animal behaviors at the GPS-determined locations (Figure 1). In the first step, we calculated the basic two metrics over two recording positions (100 s), then we extended the time interval and recalculated the metrics from 100 to 800 s. The distance moved included the cumulative distance travelled and linear distances between focal locations. Distances that occurred temporally before a considered location are called backward distances, and those after a focal location are called forward distances. The linear distance $d_{\left(b3,\;a1\right)}$ between *b3* and *a1* was calculated by Equation (1), and the $d_{\left(b1,\;b2\right)}$, $d_{\left(b2,\;b3\right)}$, $d_{\left(a3,\;a4\right)}$,$\;d_{\left(a2,\;a3\right)}$,$\;d_{\left(a1,\;a2\right)}$, $d_{1}$,$\;d_{2}$,$\;d_{3}$ and $\;d_{4}$ were used the same equation. The backward accumulative distance $d_{\left(b3,\;a1\right)}$ and the forward accumulative distance $d_{\left(a1,\;a2\right)}$ was the same as linear distance. For extending time intervals of GPS positions, the backward accumulative distance between a1 and b2 was the sum of $d_{\left(b2,\;b3\right)}$ and $d_{\left(b3,\;a1\right)}$ in Equation (2) and forward accumulative distance between a1 and a3 was the sum of $d_{\left(a1,\;a2\right)}$ and $d_{\left(a2,\;a3\right)}$ in Equation (3). For further processing of accumulative distance, the backward accumulative distance between a1 and b1 was the sum of $d_{\left(b3,\;a1\right)}$, $d_{\left(b2,\;b3\right)}$ and $d_{\left(b1,\;b2\right)}$ in Equation (4) and forward accumulative distance between a1 and a4 was the sum of $d_{\left(a1,\;a2\right)}$,$\;d_{\left(a2,\;a3\right)}$ and $d_{\left(a3,\;a4\right)}$ in Equation (5) (Figure 1). Calculation of distances metrics in other time intervals followed the same procedure. Metrics used and their meaning at time intervals of 100–800s were illustrated in Figure 2.
(1)$$d1=\sqrt{{\left(b3_{x}-a1_{x}\right)}^{2}+{\left(b3_{y}-a1_{y}\right)}^{2}}$$
(2)$$d_{\left(b1,\;a1\right)}=\;\left|d_{\left(b2,\;b3\right)}\right|+\left|d_{\left(b3,\;a1\right)}\right|$$
(3)$$d_{\left(a1,\;a3\right)}=\;\left|d_{\left(b2,\;b3\right)}\right|+\left|d_{\left(b3,\;a1\right)}\right|$$
(4)$$d_{\left(a1,\;b1\right)}=\;\left|d_{\left(b3,\;a1\right)}\right|+\left|d_{\left(b2,\;b3\right)}\right|+\left|d_{\left(b1,\;b2\right)}\right|$$
(5)$$d_{\left(a1,\;a4\right)}=\;\left|d_{\left(a1,\;a2\right)}\right|+\left|d_{\left(a2,\;a3\right)}\right|+\left|d_{\left(a3,\;a4\right)}\right|$$

Metrics of tri-axis accelerometer were calculated at 50 s intervals across the dataset of cattle, including accelerations along three orthogonal axes ($\ddot{d_{x}}$, $\ddot{d_{y}}$, and $\ddot{d_{z}}$), which was defined as three dimensional Cartesian system in neck (${\ddot{d}}_{xneck}$,$\;{\ddot{d}}_{yneck}$, and ${\ddot{d}}_{zneck}$), leg (${\ddot{d}}_{xleg}$,$\;{\ddot{d}}_{yleg}$, and ${\ddot{d}}_{zleg}$), and tail (${\ddot{d}}_{xtail}$,$\;{\ddot{d}}_{ytail}$, and ${\ddot{d}}_{ztail}$). $\ddot{d_{x}}$ is acceleration (m/s^2^) in the superiorinferior axis, $\ddot{d_{y}}$ is acceleration (m/s^2^) in the anteroposterior axis and $\ddot{d_{z}}$ is acceleration (m/s^2^) in transverse axis; Magnitude of acceleration in the neck ($M_{neck}$)was calculated by Equation (6) and $M_{leg}$ and $M_{tail}$ were calucalated by the same equation; ($SD_{x}$) standard deviation of the $\ddot{d_{x}}$ were standard deviation of $\ddot{d_{x}}$ at neck, leg, and tail calculated by Equation (7). The calculation of $SD_{y}$ and $SD_{z}$ used the same equation, $\bar{\ddot{d_{x}}}\;$ is average of $\ddot{d_{x}}$ at the neck, leg, and tail in the x-direction at the same time;
(6)$$M_{neck}=\sqrt{{\left({\ddot{d}}_{xneck}\right)}^{2}\;+{\left({\ddot{d}}_{yneck}\right)}^{2}+{\left({\ddot{d}}_{zneck}\right)}^{2}}$$
(7)$$SD_{x}=\sqrt{\frac{\sum {\left(\ddot{d_{x}}-\bar{\ddot{d_{x}}}\right)}^{2}}{n}}$$

The raw acceleration is divided into static and dynamic acceleration. The static acceleration for a focal point is average of 7 accelerations at 2.5 min. before (3 accelerations) and 2.5 min. after (3 accelerations). The dynamic acceleration was the difference between the instantaneous acceleration and the running-mean derived static acceleration [26]. Overall dynamic body acceleration (*ODBA*) at the neck, leg, or tail was the sum of absolute value of dynamic acceleration at x, y, z at the neck, leg, and tail [27]. For example, the *ODBA* at neck was calculated by Equation (8) where $A_{Xneck}$, $A_{Yneck}$, $A_{Zneck}$ are the dynamic acceleration at ${\ddot{d}}_{xneck}$,$\;{\ddot{d}}_{yneck}$, and ${\ddot{d}}_{zneck}$ at the neck. $A_{Xneck}$, $A_{Yneck}$, and $A_{Zneck}$ were calculated by Equation (9). The *ODBA* in neck ($ODBA_{head}$) was the sum of the absolute values of the dynamic accelerations from all three axes by Equation (8) and the $ODBA_{neck}$ and $ODBA_{tail}$ used the same equation. The calculation of *ODBA* for leg and tail was the same as for neck.
(8)$$ODBA_{neck}=\left|A_{Xneck}\right|+\left|A_{Yneck}\right|+\left|A_{Zneck}\right|$$
(9)$$\;A_{Xneck}=\;{\ddot{d}}_{xneck}-{\bar{\ddot{d}}}_{xneck}$$

Using the various metrics derived at intervals of 100–800 s, we built three types of models: one using GPS data-based metrics only (GPS model); another from the tri-axis accelerometer data only (tri-axis model); and a model combining the tri-axis accelerometer and GPS data-based metrics (GPS-tri model).

### 2.4. Livestock Behavior Modelling

The Random Forest algorithm classification model was used to categorize livestock behavior, with movement metrics as dependent variables and observed behaviors as independent variables [20]. Random Forest is a machine-learning algorithm that especially suits data sets with many dependent variables. Random Forest provides well-supported predictions from large numbers of dependent variables and has the ability to identify the important variables of the model [28]. The modelling process of Random Forest can be summarized as consisting of many decision trees [29]:Construct bootstrap data set (bag data set) from approximate 2/3 of the original data set; the remaining 1/3 of the data set is recognized as ‘out of bag’ (OOB).Randomly select several predictor variables to calculate nodes in the bootstrap dataset.At each decision tree node, test a random subset of predictor variables, to partition the bootstrap data into increasingly homogeneous subsets. The node-splitting variable selected from the variable subset is that which results in the greatest increase in data purity (Gini) before and after the tree node split.The trees are fully grown, and each tree is used to predict OOB data, compute accuracy, and average error rates over all predictions.The predictions are calculated by means of the majority vote of OOB predictions of the tree, and all predictions are averaged together to determine the class for the observation.

Three training parameters need to be defined in the Random Forest algorithm; these parameters then determine the model prediction power:

Our analysis is carried out with the caret package in R Studio (R Development Core Team 2011) by using the Random Forest, caret, and plotmo packages. When building Random Forest models within this package there are two main user-controlled parameters: the number of variables to try at each node (the ‘mtry’ argument), and the number of trees in the forest (the ‘ntree’ argument). We used the train() function from the caret package to get an optimal combination of ‘mtry’ and ‘ntree’. The train() function was run for 10 (‘mtry’ from 1 to 10) times. To determine the optimal number of trees for our data, the approach was to create many ‘caret’ models for our algorithm and pass in a different value of ‘ntree’ while holding ‘mtry’ constant at the default value above. We tested models with varying numbers of trees as a function of tree number of tress approaches a flat line between 500 and 2000 trees.

Mean decrease in Gini is used to determine the importance of variables in the classification model; this parameter is based on the Gini impurity index used for the calculation of splits during training [20]. When a tree is built, the decision regarding which variable to split at each node uses the Gini parameter. For each variable, the sum of the Gini decrease across every tree of the forest is accumulated every time that variable is chosen to split a node. The sum is divided by the number of trees in the forest to give the mean decrease in Gini.

### 2.5. Performance of the Random Forest Classifier

The performance of Random Forest classification models was evaluated by using two indices: overall accuracy and the κ coefficient [30]. Overall accuracy represents the proportion of the total number of correctly classified observations. The κ coefficient, which considers the agreement occurring by chance, is a statistical measure of inter-rater agreement for categorical items [30].

To evaluate the performance of the Random Forest model, we used 10-fold (i.e., performed 5 times) cross-validation to separate the data set into different, smaller data sets as training data sets and testing data sets. This process enabled us to more precisely control the number of samples compared with the inherent bootstrap sample in the Random Forest model [31].

## 3. Results

### 3.1. Performance of GPS, Tri-Axis, and GPS-Tri Axis Models

Overall classification accuracy increased as the time interval increased: 84.4%, 84.5%, 86.44%, and 87.6% at time intervals of 100, 150, 200, and 250 s. For all GPS models, accuracy began to plateau around 0.89–0.91, when the time interval was greater than 300–800 s. For both the GPS-tri and tri-axis models, overall classification accuracy was approximately 96% at all time intervals (Figure 2).

Compared with the relatively small change in overall classification accuracy with different time intervals, the κ coefficient for GPS models increased dramatically from 7% to 42% as the time interval increased from 100 to 250 s. The κ coefficient stabilized at 57% to 65% when the time interval exceeded 300 s (Figure 2). The GPS-tri and tri-axis models yielded approximately the same κ coefficient (91% to 92%, 92%) at all time intervals (Figure 3).

### 3.2. Cross-Validation

For GPS models with time intervals of 100 to 800 s, the accuracy for grazing behavior was 92% to 98%, whereas the accuracy for nongrazing behavior increased from 20% to 47% as the time interval increased from 100 to 250 s and from 58% to 66% with time intervals of 300–800 s (Table 2). The performances of tri-axis were showed accuracy for grazing behaviors (98%) and nongrazing (92%) (Table 3).

### 3.3. Relative Importance of Variables

The first four metrics in order of importance (as indicated by the mean decrease in Gini) for the GPS model with time intervals from 100 to 800 s are shown in Figure 3 and Appendix A. In most of the models, either linear or accumulated distance, rather than turning angle, was the important metric in the modelling. The time lag until the important distance metric occurred increased with the time interval from 100 to 800 s (Figure 4). Among all of the important metrics at different time intervals, *d*19 (the backward linear distance at a time interval of 300 s) and *d*43 (backward linear distance at a time interval of 350 s) were the most frequently used metrics in the classification of livestock behaviors. The variable *d*19 was the most important for the GPS models when the time interval was 300–600 s, and *d*43 was most important for time intervals from 350 to 700 s.

In the tri-axis model, the variable ${\ddot{d}}_{yneck}$(acceleration of anterior–posterior movement in the neck) had the highest mean decrease in Gini, and $M_{tail}$ (square root mean of the sum of acceleration in the neck, leg, and tail) the second largest. The mean decrease in Gini gradually declined from ${\ddot{d}}_{yleg}\;$(acceleration of anterior–posterior movement in the foot) to ${\ddot{d}}_{xleg}$ (acceleration of superior-inferior movement in the foot) but then dramatically decreased from ${\ddot{d}}_{xleg}$ to ${\ddot{d}}_{zneck}$ (acceleration of transverse movement in the neck) (Figure 5).

### 3.4. Marginal Effect of the Variable on Livestock Behavior Classification

We used partial dependence plots to show the marginal effect of the metrics used in the behavior classification. For all GPS models, we generated partial dependence plots for the first four most important variables determined according to the mean decrease in Gini (Figure 2).

Although *d*19 and *d*43 had important roles in behavior modeling, the marginal probability of classifying a behavior as nongrazing decreased as the time interval increased. The probability of nongrazing showed a sharp decrease when *d*19 and *d*43 were greater than approximately 35–50 m. In the GPS model at the 300 s time interval, the marginal probability to classify a behavior as nongrazing was around 0.4 when *d*19, *d*18 (the backward linear distance at a time interval of 250 s), *d*17 (the backward linear distance at a time interval of 200 s), and *d*20 (the backward accumulative distance at a time interval of 200 s) were less than 35–50 m (Figure 6A), thus accounting for more than 80% of the total behavior in this range of distance (Figure 6B). The utility power of these four distances in classifying a behavior as nongrazing gradually decreased and then stabilized around 0.22 when they were greater than 50 m (Figure 6A).

In the tri-axis model, when ${\ddot{d}}_{yneck}$ was less than −3 m/s^2^, the behavior was never classified as nongrazing, whereas the probability of a behavior being classified as nongrazing was around 0.8 when $\;{\ddot{d}}_{yneck}$ was greater than −3 m/s^2^. For the variable $M_{tail}$, the probability of a behavior being classified as nongrazing was 0.5 when $M_{tail}$ was 0 m/s^2^ and dropped dramatically to 0.3 when $\;M_{tail}$ was 7 m/s^2^. The behavior being classified as nongrazing was 0.3 when ${\ddot{d}}_{yleg}$ was from −20 to 0 m/s^2^, dropped to 0.22 when ${\ddot{d}}_{yleg}$ was 8 m/s^2^, increased to 0.25 when ${\ddot{d}}_{yleg}$ was more than 11 m/s^2^. By using ${\ddot{d}}_{xleg}$, the highest marginal probability of determining a behavior as nongrazing was 0.31 and dropped to 0 when $\;{\ddot{d}}_{xleg}\;$ was 11 m/s^2^ (Figure 7).

## 4. Discussion

### 4.1. Optimal Time Interval for GPS Models

GPS location data can be used to infer latent states of behavior from within individual movement trajectories [19]. The duration to complete a specific behavioral activity depends on the type of livestock and the condition of the pasture [6]. Distance and turning angle metrics extracted from GPS data over specific time intervals can be used to classify livestock behaviors, such as 1 min for beef cows on desert grassland [6], 3 min for Brown Swiss cows in a cow shed [11], and 5 min (i.e., 300 s) for dairy cows on upland grassland [19]. In our study, the optimal time interval for behavior classification was approximately 300 s because the κ coefficient at this time interval was higher than for shorter time intervals and was nearly stable afterward (Figure 3). In addition, the most frequently used metric (*d*19) was the backward linear distance at the 300 s time interval (Figure 4).

Although overall accuracy did not vary over time intervals from 100 to 800 s, it may be a poor measure for assessing model performance, given that overall accuracy can happen just due to coincidence, especially when the data are imbalanced [6]. In contrast, the κ coefficient, which estimates accuracy beyond expectation, can correctly assess the accuracy of imbalanced data [32]. For imbalanced data, the observed and predicted accuracies and their agreement in regard to minor behaviors determine the κ coefficient. In reality, foraging occurs more often than other behaviors. During the cross-validation, given that the accuracies for grazing behavior were relatively high and stable, the critical determinants of the κ coefficient were the accuracies for nongrazing behaviors. For the GPS models, the low accuracies of the nongrazing behaviors during cross-validation (Table 2) explain the low κ coefficients for the time intervals from 100 to 250 s (Figure 3). At time intervals of 300 s and greater, the κ coefficient stabilized around 0.5–0.6 because of the increase in the accuracies of nongrazing behavior (Table 2). In addition, the d19 (backward linear distance at 300 s) was the most frequent metric in other models when the time interval was greater than 300 s (Figure 4). Therefore, the optimal time interval for using the GPS location data to classify the livestock behavior in the study area was 300 s.

### 4.2. Model Performance

Predicting the accuracy of models by using GPS data depends on the livestock type and the pasture condition [21], but when using tri-axis accelerometer data it depends only on the instantaneous body posture of the animal [15]. With the same time step to log the GPS position and the body posture by tri-axis accelerometer, models using tri-axis accelerometer data-based metrics only or combined tri-axis and GPS data-based metrics showed higher overall accuracies and κ coefficients than the models that used only GPS data-based metrics (Figure 3).

The distance moved by a livestock over a given time interval is expected to be an indicator of its activity. Short distances are likely to indicate static behavior (standing, ruminating), and long distances typically are associated with foraging [33]. In the current study, distance variables were the first four most important variables in most of the GPS models (Figure 4), thus supporting the power of using distance to classify cattle behavior.

The GPS models demonstrated several critical distances for classifying grazing and nongrazing behaviors (Figure 4). But, the marginal probabilities of the important variables to distinguish between grazing and nongrazing behaviors were lower for the GPS models than for the tri-axis models (Appendix A and Figure 7). Moreover, the distances tended to be within the range that ambiguously classified the two behaviors (Appendix A). Therefore, distinguishing between grazing and nongrazing was particularly challenging and relied on the use of multiple movement metrics, including backward and forward linear and accumulative distances (Figure 4). For example, for the 300 s time interval, *d*19 was the first most important metric to determine the two behaviors. The marginal probability for nongrazing was approximately 40%, meaning unclear differentiation between grazing and nongrazing when *d*19 was less than 35 m. However, the probability of nongrazing was around 20%, indicating that the two behaviors were clearly differentiated when *d*19 exceeded 35 m. Unclear classification at shorter distances than this critical distance (35 m) might reflect the condition of the specific habitat. For example, the presence of woody vegetation might have made it more difficult to distinguish between grazing and nongrazing, because the consumption of shrubs slows movement and can blur the graze signature in terms of the motion sensor counts. In addition, 89% of the d19 data were less than 35 m. Hence, the lower probability of the distance metrics to classify the two behaviors under the threshold value and the skewed distribution of these metrics could be responsible for the relatively low accuracy of the GPS models.

The tri-axis accelerometer model was based on the body posture that was simultaneously associated with a specific behavior and did not need to account for any time interval, which might lead to uncertainty regarding behavior classification [34]. Unlike the GPS model, the tri-axis model can measure the instantaneous and independent local movement of the legs, heads, or entire bodies of animals, thus ensuring high accuracy of behavior classification [15,16,17,18]. Our findings showed that the anteroposterior movement of the neck was critical for distinguishing livestock behaviors (Figure 5), in agreement with the results of another study, which used x-axis sensor counts [14].

Livestock behaviors were influenced by the available forage and stocking density. With increasing stocking density, the average intake of each livestock will reduce due to the given availability forage in the rangeland [35]. Livestock preferred to spend less time on grazing behaviors when consuming of energy was more than grain [35]. More available forage in August (243 g/m^2^) than that in July (53 g/m^2^) in Horqin Sandy Land might lead to the livestock spending more time on grazing with sufficient energy of forage in August. For the behavior’s classification, livestock may spend less time over a given distance for finishing grazing behavior. So, the optimal time-interval of the GPS method for classifying behaviors will decrease. Our GSP model was built over 100–800 s to cover various situations corresponding with the change of rangeland pasture, thus the method can be applied in other sites. 

## 5. Conclusions

Our current study demonstrates that data from both GPS devices and tri-axis accelerometer can be applied to build reliable models for livestock behavior classification.

To achieve the high and stable performance of the GPS model, we selected the optimal time interval from 300 to 800s, which is sufficient for most livestock activities associated with behaviors to be displayed. Metrics of linear distance had the most important effects on behavior classification. In addition, the marginal effects of linear distance indicated a distance of 35–50 m was the threshold for differentiating behaviors. At longer distances, grazing was more likely than nongrazing behavior.

Because it is based on the instantaneous acceleration of livestock body movement, the tri-axis model achieves higher performance regarding livestock behavior classification than does the GPS model. The anteroposterior movement of the animal’s neck was the most important metric for the tri-axis model. The marginal effects showed that acceleration of −3 m/s^2^ was the threshold for differentiation of behaviors; at greater values, nongrazing was more likely than grazing.

In summary, compared with GPS models, a tri-axis model can better support livestock behavior classification, which is advantageous for assessing the detailed activities associated with investigating livestock physiology. But the main disadvantage of a tri-axis model is its lack of location information. A GPS model is sufficient for livestock behaviors classification and provides information regarding an animal’s location; this feature is associated with the interaction between livestock activities and the rangeland ecosystem. These findings may improve our understanding of how the selection of the time interval influences the process of distinguishing livestock activities in a GPS model and provide insight into selecting an optimal time interval when using GPS data only to classify livestock behaviors. 

## Figures and Tables

**Figure 1 sensors-19-05334-f001:**
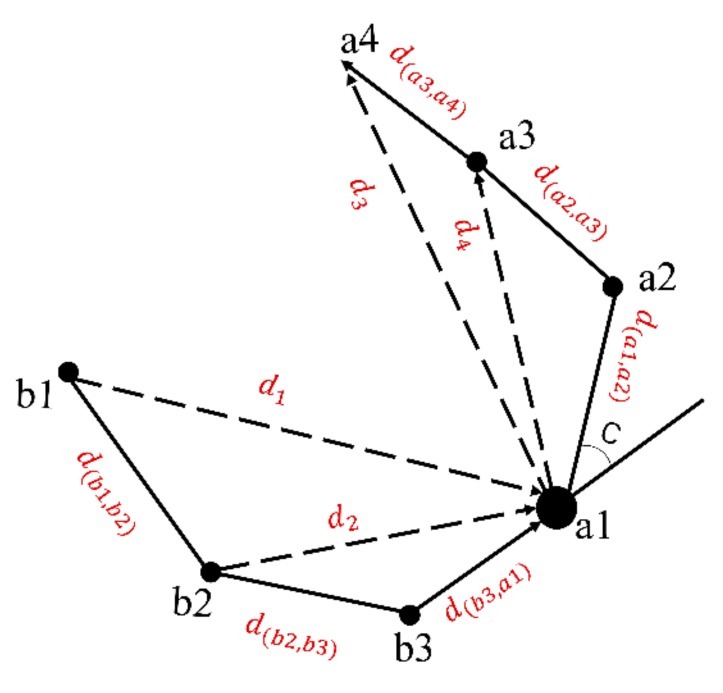
Schematic representation of movement metrics used as predictive metric in the classification. Movement metrics include backward accumulative distance ($d_{\left(b2,\;a1\right)}$, $d_{\left(a1,\;b1\right)}$), forward accumulative distance ($d_{\left(a1,\;a3\right)}$, $d_{\left(a1,\;a4\right)}),\;$ backward linear distance ($d_{1}$, $d_{2}$), forward linear distance ($d_{3}$, $d_{4}$), and turning angle between Global Positioning System (GPS) positions (c).

**Figure 2 sensors-19-05334-f002:**
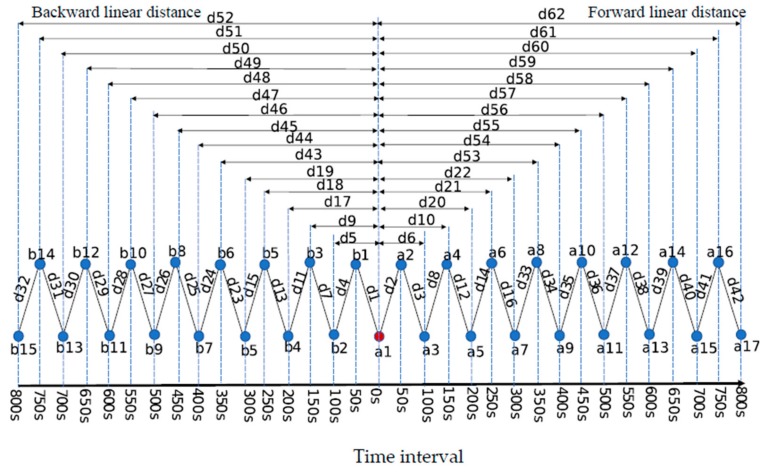
Metrics of distance extracted from GPS device were used to classify livestock behaviors from 100 to 800s time intervals in Random Forest model. a_1_ is the focal point, *a*_2-17_ and *b*_1-16_ were forward and backward locations at time interval from 100 to 800s. *d*_1_-*d*_62_ were the forward and backward linear distance metrics of distance from 100 to 800s time interval. Accumulative distances were calculated by Equations (2)–(5). *d*_62_-*d*_92_ were the accumulative distances metrics used in the model. Forward accumulative distance: *d*63 = *d*2 + *d*3; *d*64 = *d*63 + *d*8; *d*65 = *d*64 + *d*12; *d*66 = *d*65 + *d*14; *d*67 = *d*66 + *d*16; *d*68 = *d*67 + *d*23; *d*69 = *d*68 + *d*34; *d*70 = *d*69 + *d*35; *d*71 = *d*70 + *d*36; *d*72 = *d*71 + *d*37; *d*73 = *d*72 + *d*38; *d*74 = *d*73 + *d*39; *d*75 = *d*74 + *d*39; *d*76 = *d*75 + *d*41; *d*77 = *d*76 + *d*42. Backward accumulative distance: *d*78 = *d*1 + *d*4; *d*79 = *d*78 + *d*7; *d*80 = *d*79 + *d*11; *d*81 = *d*80 + *d*13; *d*82 = *d*81 + *d*15; *d*83 = *d*82 + *d*23; *d*84 = *d*83 + *d*24; *d*85 = *d*84 + *d*25; *d*86 = *d*85 + *d*26; *d*87 = *d*86 + *d*27; *d*88 = *d*87 + *d*28; *d*89 = *d*88 + *d*29; *d*90 = *d*89 + *d*30; *d*91 = *d*90 + *d*31; *d*92 = *d*91 + *d*32. The meaning and time interval of a specific accumulative distance can be read from Figure 2. For example, *d*63 = *d*2 + *d*3, thus *d*63 is the forward accumulative distance at 100s.

**Figure 3 sensors-19-05334-f003:**
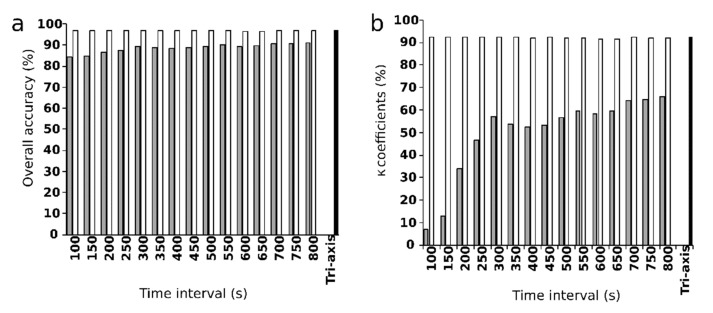
(**a**) Overall accuracy and (**b**) κ coefficients of the GPS (gray bars) and GPS-tri (white bars) with time intervals of 100–800 s and of the tri-axis model (black bars).

**Figure 4 sensors-19-05334-f004:**
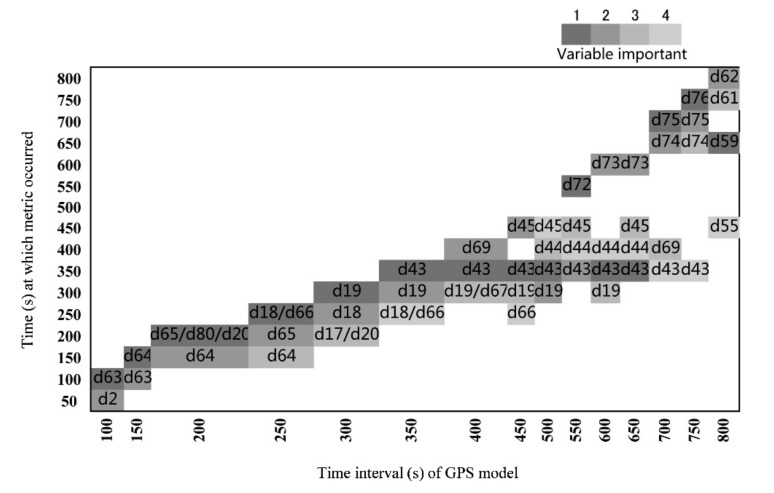
Variable importance plot generated by using the Random Forest algorithm with GPS models. The plot shows the first four important metrics of each GPS model (1, 2, 3, 4) according to the mean decrease in Gini; as this parameter increases, the variable is more important and a more accurate predictor of behavior classification. See Figure 2 and equation (Equations (1)–(6)) for the meaning of metrics.

**Figure 5 sensors-19-05334-f005:**
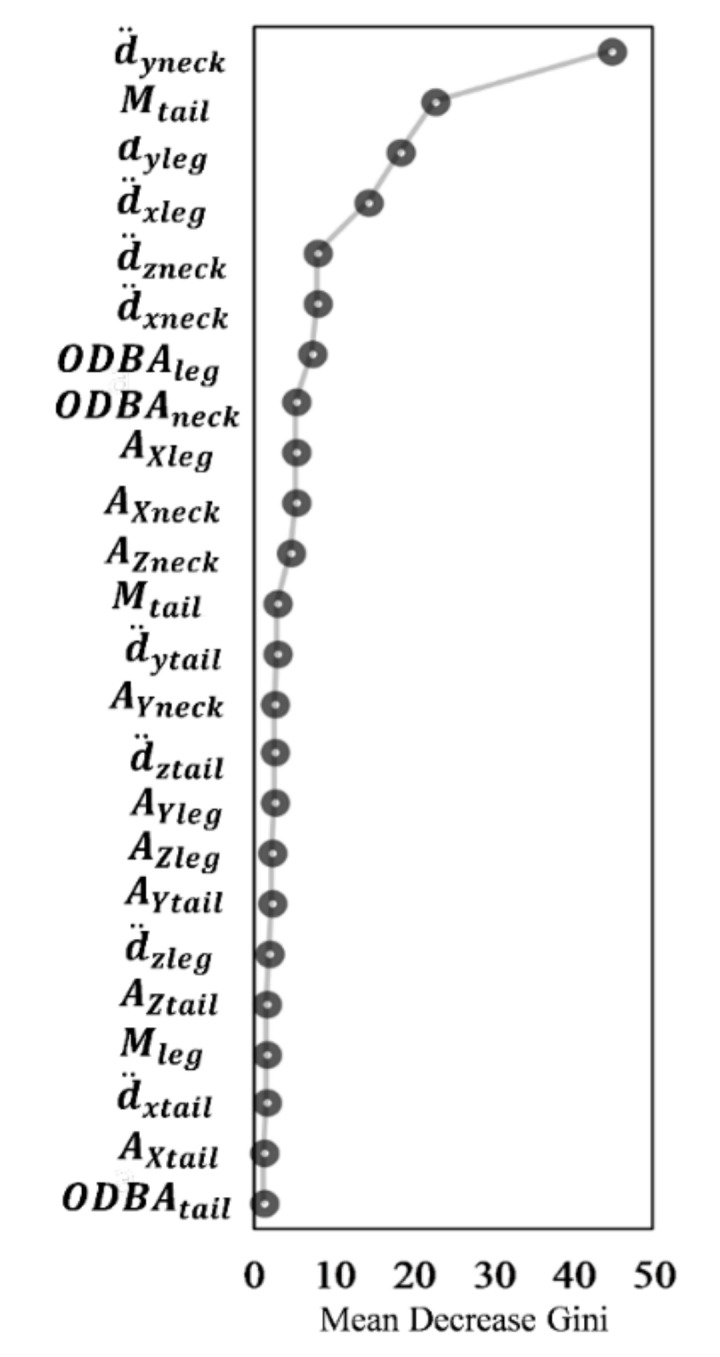
Variable importance plot generated by using the Random Forest algorithm with the tri-axis model. The plot shows the importance of each variable according to the mean decrease in Gini; as this parameter increases, the variable is more important and a more accurate predictor of behavior classification. See Equations (6)–(9).

**Figure 6 sensors-19-05334-f006:**
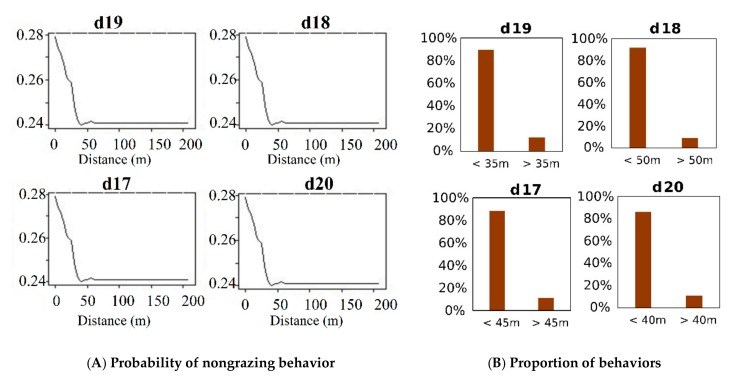
Partial dependence plots of nongrazing (**A**) and the proportion of behaviors corresponding to threshold in the GPS model (**B**). Partial plots represent the marginal effect of a single metric (*d*19, *d*18, *d*17, *d*20) of 300 s time-interval included in the Random Forest model on the probability of nongrazing behavior, when the effects of all other metrics are averaged out. The criteria of threshold distance of each partial plot are recognized that the nongrazing behaviors remain same probability. See Figure 2 and Equations (6)–(9) for the meaning of metrics.

**Figure 7 sensors-19-05334-f007:**
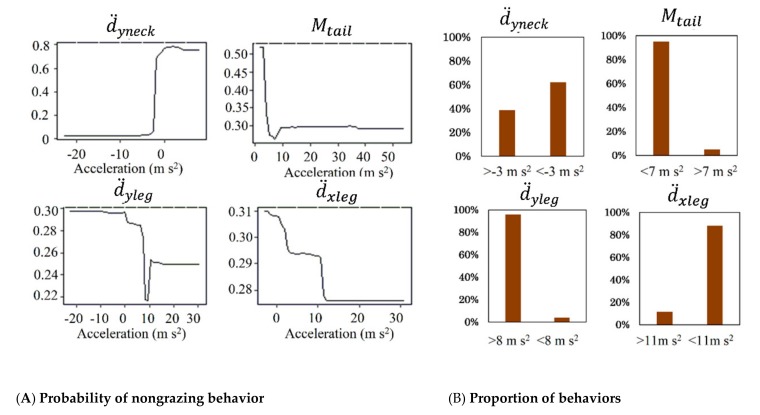
Partial dependence plots of nongrazing (**A**) and the proportion of behaviors corresponding to threshold in the tri-axis model (**B**). Partial plots represent the marginal effect of a single metric (${\ddot{d}}_{yneck,\;}\;M_{tail,\;}{\ddot{d}}_{yleg},\;\;{\ddot{d}}_{xleg}$) included in the Random Forest model on the probability of nongrazing behavior, when the effects of all other metrics are averaged out. The criteria of threshold distance of each partial plot are recognized that the nongrazing behaviors remain same probability. See Equations (6)–(9) for the meaning of metrics.

**Table 1 sensors-19-05334-t001:** Descriptions of the observed behaviors (modified from Ganskopp and Bohnert [12]).

Behavior category	Definition	Explanation
Grazing	Foraging, Foraging–walking	Foraging: foraging continuously (head lowered) Foraging–walking: foraging while walking (head raised and lowered)
Nongrazing	Standing, Lying down, Rumination	Standing: the animal stands on all four legs, with head erect and without swinging its head from side to side Lying down: the cattle lies on the ground in any position (except flat on its side) without ruminating Ruminating: the cattle lies in a stall masticating regurgitated feed, swallowing masticated feed, or regurgitating feed with head erect

**Table 2 sensors-19-05334-t002:** The confusion matrix for livestock behaviors classification as categorized by using GPS models with time intervals of 100–800 s.

Observed Behaviors	Predicted Behaviors								
	Grazing	Nongrazing	Percent Accuracy	Grazing	Nongrazing	Percent Accuracy	Grazing	Nongrazing	Percent Accuracy
	100 s			150 s			200 s		
Grazing	421	35	0.92	428	28	0.94	428	28	0.94
Nongrazing	66	17	0.20	63	20	0.24	51	32	0.39
	**250 s**			**300 s**			**350** s		
Grazing	427	29	0.94	430	26	0.94	433	23	0.95
Nongrazing	44	39	0.47	30	53	0.64	34	49	0.59
	**400 s**			**450s**			**500 s**		
Grazing	447	9	0.98	440	16	0.96	446	10	0.98
Nongrazing	33	50	0.60	31	52	52	35	48	0.58
	**550 s**			**600 s**			**650 s**		
Grazing	446	10	0.98	444	12	0.97	445	11	0.98
Nongrazing	35	48	0.59	33	50	0.6	32	51	0.61
	**700 s**			**750 s**			**800 s**		
Grazing	442	14	0.97	440	15	0.96	435	21	0.95
Nongrazing	32	51	0.61	28	55	0.66	29	56	0.66

For each row, accuracy was calculated as the proportion of the observed class relative to the total number of behaviors.

**Table 3 sensors-19-05334-t003:** The confusion matrix for livestock behaviors classification as categorized by using the tri-axis model.

Observed Behaviors	Predicted Behaviors		
	Grazing	Nongrazing	Accuracy
Grazing	447	9	0.98
Nongrazing	7	76	0.92

For each row, accuracy was calculated as the proportion of the observed class relative to the total number of behaviors.

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
