# Peer review of "Method for Classifying Behavior of Livestock on Fenced Temperate Rangeland in Northern China"

_sensors, 2019, doi:10.3390/s19235334_

Round 1
Reviewer 1 Report
This paper addresses a question that is appropriate for this Journal. However, there are numerous issues to correct before a thorough and critical review can be conducted. I have listed my concerns with the paper below, but the paper is missing information that is needed to conduct a critical review.
The introduction needs to be revised. Spatial grazing distribution and overgrazing are critical issues for rangeland management. However, the introduction does not justify why determination of behavioral activity is needed to address management of spatial grazing distribution. I believe there are reasons, but the paper does not list them. My suggestion is to talk about the need to separate cattle locations into areas where cattle rest and areas where they graze.
Line 51 is a terrible introductory sentence for this paragraph. I suggest deleting this sentence.
Line 53. Replace “of rangeland” with “of forage species”
Lines 64-68. Talk about accelerometers and being able to remotely monitor animal motion before talking about statistical models.
Lines 11-120. The description of behavioral observation is not adequate. How were behavioral observations recorded? Continuously? What was used to record the time of visual observations? How many observers were there? How close did the observers get to the cattle? How did they handle transitions in behavior? When analyses of 100 to 800 seconds were conducted, how were the visual observations change from one time interval to another?
Lines 123-128. The methods used to derive movement metrics is confusing and unclear.
Table 2 is completely worthless. The values listed in the table have no meaning to me, and I use random forest procedures to predict livestock behavior from GPS tracking and accelerometers. The caption for Table 2 is does not provide any indication how to interpret and use the values in the table.
Line 139-140. The calculation of static and dynamic acceleration needs further description, not just one sentence. For example, if the static component was calculated over 300 seconds how was the dynamic component calculated for the time intervals of 100 to 800 seconds. Methodology is missing here.
Table 3. Most papers use equations to describe calculated metrics from accelerometer data. At the very least, references and brief explanations should be given. What does magnitude and SMA mean here? There are several way to calculate magnitude.
Table 4 is worthless. It only refers to other tables.
Lines 166-175. The authors state that certain parameters used in the Random Forest algorithm gave the highest accuracies. How was this determined? What metric was used to determine accuracy here? Again more information in the Methods is needed.
Although cross-validation is useful for validation, a more robust approach was to test the final model against a completely independent data that was absolutely independent from the training data. I expect this was not done because there was not sufficient visual observation data collected. The visual observation data in this study is minimal!
Table 5 is a confusion matrix. The table caption and headings are not correct. See the following site for help with confusion matrix terminology: https://www.dataschool.io/simple-guide-to-confusion-matrix-terminology/
Table 6 is also a confusion matrix and the terminology is wrong in this table as well. The terms “sensitivity” and “specificity” are often used.
Figure 2 is not useful. The values in the graph or not helpful (e.g., “d64”)
Figures 3, 4 and 5 are not useful. Acronyms used are not defined.
Lines 244 to 256 are not useful. The acronyms used in this paragraph are not defined or helpful.
Lines 345-352. The rationale for deriving cattle behavior from GPS and/or accelerometers to manage spatial grazing issues is not coherent and the rationale is missing. The conceptual framework for this research needs more work and explanation.
Conclusions. The authors appear to want to use 300 seconds over 800 seconds because there is really no difference among time intervals between 300 and 800 seconds. Accuracy declines with shorter time intervals the 300 seconds. The authors are missing an important point. For spatial grazing studies, longer time intervals (800 seconds rather than 300 seconds) because longer intervals between positions allows the GPS to track the cattle longer which is much better for describing movement patterns. Cattle movement patterns vary from day to day.
Author Response
Dear Reviewer,
Thank you for your letter and the comments concerning our manuscript entitled ‘Method for classifying behavior of livestock on fenced temperate rangeland in northern China’ (Manuscript ID: sensors-592530). Those comments are valuable and very helpful for revising and improving our manuscript, as well as the important guiding significance to our researchers. We have carefully studied comments and made corrections which we hope meet with approval. The revised portion is marked in red in the paper.
Please note that the comments from reviewers are in italic followed by our response in regular text.
Please see the attachment.
Best regards,
Peng Fei

Reviewer 2 Report
With this paper the authors describe their use of a machine learning approach to process GPS positional data and tri-axis accelerometer data to classify cattle behavior as grazing or non-grazing. The data collected were sufficient to predict cattle behavior with high accuracy, and the tri-axis accelerometer data provided a significant increase in behavior classification accuracy over positional data alone. The methods and results are described in sufficient detail and the paper is well written. A few specific suggestions given below could improve the clarity and interpretation of the presentation of results. Suggestions: Abstract: The abstract is around 328 words, considerably longer than the 200 word maximum given in instructions to authors. The abstract could be clarified with a statement to the effect that the authors compared predictive models using GPS data only, tri-axis accelerometer data only, and GPS and accelerometer data combined (e.g., briefly explain the terms "GPS model", "GPS-tri model", etc). Otherwise, the abstract could be improved by omitting some detail in terms of model performance. Methods: I would expect that relative availability of forage would influence the movement rate and grazing rate of the cattle: with low forage availability per animal, the animals would need to move more to reach food (see Hepworth et al. 1991 (Journal of Range Management, "Grazing Systems, Stocking Rates, and Cattle Behavior in Southeastern Wyoming")). So, it seems that this would influence the relationship between sensor data and behavior states. Could use a little more background on the condition of the pasture where animals were grazing, to inform interpretation of how relevant this work may be to new systems. Please add some information to the methods section describing general forage availability relative to the density of animals in the pasture. If possible, describe forage availability relative to the stocking rate of animals in the pasture, for example with reference to animal unit equivalents (see Scarnecchia 1985, Rangeland Ecology and Management: "The animal-unit and animal-unit-equivalent concepts in range science"). Results: Figure 1a is missing its y-axis label. I am confused by the use of both "observed accuracy" and "predicted accuracy" in table 5 and in the text. Please add an explanation to the text as to why these two metrics are needed. Why is it not sufficient to report the percent of cases where the predicted classification of behavior matched the observed classification? Does figure 4 pertain to the 300-s time interval? if so, please include in the figure caption. Discussion: If livestock spend ~80% of their time grazing (as detected in the current study), what is the value of discriminating grazing vs non-grazing behavior? GPS location technology would be enough to capture impacts of grazing. I'm not yet convinced that location alone does not primarily drive impacts. As mentioned above with reference to the methods section, please add a brief discussion of the forage availability relative to stocking density in the pasture being grazed. Could the findings of the current study be cautiously extended to other locations? what site characteristics would limit the extension of the current findings to new locations? References: The references in literature cited do not match the format required by the journal. SI: The format of figure 1 in the SI makes the figure difficult to read. Please rotate the y-axis labels so that they are oriented the same way for both probability, and time interval. Also, there appear to be several rows of the figure missing, corresponding to time intervals of 100, 150, 200, and 300 seconds. Please add these rows or add a statement to the figure caption in SI as to why they were omitted.Author Response
Dear Reviewer:
Thank you for your letter and the comments concerning our manuscript entitled ‘Method for classifying behavior of livestock on fenced temperate rangeland in northern China’ (Manuscript ID: sensors-592530). Those comments are valuable and very helpful for revising and improving our paper, as well as the important guiding significance to our researchers. We have carefully studied comments and made corrections which we hope meet with approval. The revised portion is marked in red in the paper.
Please note that the comments from reviewers are in italic followed by our response in regular text.
Best regards,
Peng Fei

Round 2
Reviewer 1 Report
Lines 54-58. The authors attempted to explain the need to classify livestock behavior to understand livestock spatial distribution and its impacts to pastures and management, but the argument is flawed. Harris et al. (2003) citation 10 does not provide any insight into the impact of non-grazing behaviors. Harris et al. (2003) only compares an area grazed for years to an area that has not been grazed for years. The authors need a new and appropriate citation to support this sentence.
Lines 59-65. To use labor demand as a rationale for using accelerometers and GPS to classify behavior is not really appropriate unless you at least briefly discuss how data processing can be automated so that farm managers do not have to use sophisticated statistical analyses and data processing to use the accelerometer and GPS tracking data.
Lines 94-95. Something needs to be added before “[24]”, the citation. Animal unit month is a commonly used metric for stocking rate (see {24]). The value given “1.54 cattle/ha” is not an appropriate value for stocking rate. Instead it is a value for stocking density. Stocking rate must have a time element (see the {24} reference). Based on the information, given the stocking rate in this study was 0.51 Animal Unit Months per hectare.
Lines 113-124. The explanation of behavioral data collection is improved. However, it is unclear how many cattle were observed. As written, the observer only collected data on 1 of the 13 cows. If so, the behavioral data may be biased. There can be differences in accelerometers and the accuracy of GPS units can vary.
Line 144. The calculation of ODBA is not explained here. The explanation of ODBA is discussed below.
Lines 144-147. The explanations here are confusing. If I understand this, it would be clearer to delete “six recording positions of GPS (n=6) and add “the previous” before “5 min”. X and M should also be defined. Was SD only calculated for superior-inferior axis.
Tables 3 and 4. The authors did not correctly label confusion matrices and did not use the information that I provided them for correctly labeling confusion matrices. The problem is that you cannot have 4 different accuracies in this table. Accuracy is not the correct terminology for both the two rows and two columns. See
https://www.dataschool.io/simple-guide-to-confusion-matrix-terminology/
The authors did not solve the problem with Figures 3, 4, 5 and 6. The figures do not have sufficient explanation to be useful. Values such as “d2” or “d65” in Figure 3 only have meaning to the authors and mean nothing to the reader. Similarly “M2” “Y1”…. Are not defined in Figure 4, and it is up to the reader to guess what they mean. In Figure 5, the authors assume the reader knows what “d17 “d18” “d19: or “d20” means. Without knowing what the metric is the figure is worthless.
Lines 507-514. This paragraph is confusing and difficult to read. I am not sure what the authors were trying to discuss, but it does not help me understand how the GSP model built over 100 to 800 seconds can be applied to other sites. I suggest deleting this paragraph.
Conclusions. The rationale for suggesting 300 seconds as the optimal time interval is not sufficiently justified.
Author Response
Dear Reviewer,
Thank you for your nice comments concerning our manuscript entitled ‘Method for classifying behavior of livestock on fenced temperate rangeland in northern China’ (Manuscript ID: sensors-592530).
Those comments are valuable and very helpful for revising and improving our manuscript, as well as the important guiding significance to our researchers. We have carefully studied comments and made corrections which we hope meet with approval. The revised portion is marked in red in the paper.
Please note that the comments from reviewers are in italic followed by our response in regular text.
Please see the attachment.
Best regards,
Peng Fei
